# KÉPLET: Knowledge-Enhanced Pretrained Language Model with Topic Entity Awareness

**Yichuan Li**[*]
Worcester Polytechnic Institute
yli29@wpi.edu

**Jialong Han**[†]
Airbnb Inc.
jialonghan@gmail.com

**Kyumin Lee**
Worcester Polytechnic Institute
kmlee@wpi.edu

**Chengyuan Ma** and **Benjamin Yao** and **Derek Liu**
Amazon Alexa AI
chengyuan.ma@gmail.com, benjamy@amazon.com, derecliu@amazon.com

## Abstract

In recent years, Pre-trained Language Models (PLMs) have shown their superiority by pre-training on unstructured text corpus and then fine-tuning on downstream tasks. On entity-rich textual resources like Wikipedia, Knowledge-Enhanced PLMs (KEPLMs) incorporate the interactions between tokens and *mentioned entities* in pre-training, and are thus more effective on entity-centric tasks such as entity linking and relation classification. Although exploiting Wikipedia's rich structures to some extent, conventional KEPLMs still neglect a unique layout of the corpus where each Wikipedia page is around a topic entity (identified by the page URL and shown in the page title). In this paper, we demonstrate that KE-PLMs without incorporating the topic entities will lead to *insufficient entity interaction* and *biased (relation) word semantics*. We thus propose KÉPLET, a novel **K**nowledge-**É**nhanced **P**re-trained **L**anguag**E** model with **T**opic entity awareness. In an end-to-end manner, KÉPLET identifies where to add the topic entity's information in a Wikipedia sentence, fuses such information into token and mentioned entities representations, and supervises the network learning, through which it takes topic entities back into consideration. Experiments demonstrated the generality and superiority of KÉPLET which was applied to two representative KEPLMs, achieving significant improvements on four entity-centric tasks[1].

## 1 Introduction

*Pre-trained language models* (PLMs) (Radford et al., 2018; Devlin et al., 2019; Liu et al., 2019) have shown their effectiveness on many natural language understanding tasks. To exploit the rich syntactic and semantic information in the pre-training

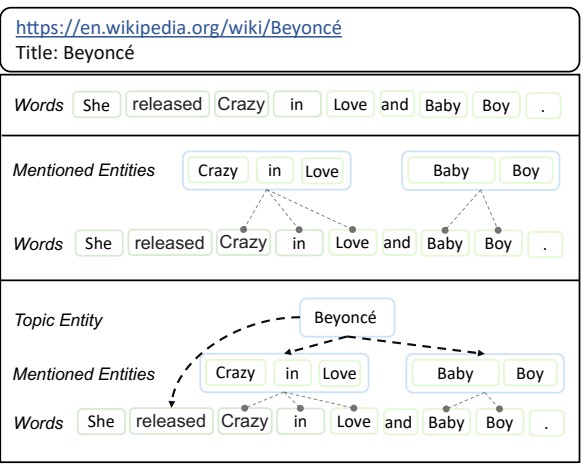

Figure 1: An illustration of Wikipedia page Beyoncé. There are three levels of interactions among words, mentioned entities and topic entity. We use ➝● to stand for the entity and words linkage, and → to express an entity interaction need to be considered and word semantics need to be covered by modeling the *topic entity*.

data, PLMs are designed to model the word co-occurrences as shown at the top of Fig. 1. However, they usually fall short in discovering factual knowledge (Logan et al., 2019) and applying such knowledge in language understanding (Zhang et al., 2019). For example, in sentence *"She released Crazy in Love"* on the Wikipedia page of Beyoncé, PLMs will try to mask and predict words like *"Crazy"* and *"in"*, not knowing that Crazy in Love is a *mentioned entity* but tearing it apart. To incorporate entity knowledge into PLMs, *knowledge-enhanced* PLMs (KEPLMs; Zhang et al.; Yamada et al.; Qin et al.; Liu et al.) are proposed to work on not only word co-occurrences but also interactions between words and mentioned entities as well as among the latter. As shown in the middle of Fig. 1, most KEPLMs work on entity-rich textual resources like Wikipedia, and consider hyperlinks of a Wikipedia page as mentions of the target pages' entities to model the rich interactions. Auxiliary objective functions are also designed to enforce the

---

[*] Work performed during internship at Amazon.
[†] Work done when Jialong was with Amazon.
[1] Code is available at https://github.com/bigheiniu/KEPLET.

model to learn entity knowledge from interactions. In general, KEPLMs have achieved superior performance compared with PLMs in entity-centric tasks (Safavi and Koutra, 2021; Yin et al., 2022), like named entity recognition, entity typing *etc.*

KEPLMs have gained their efficacy by modeling entities mentioned in Wikipedia-like textual corpus. Unfortunately, they still did not fully exploit the entity knowledge there, in the sense that they neglected another important role entities named, *topic entities*, which are page titles in the corpus. For example, in Fig. 1, those sentences are actually from the page of Beyoncé and around this celebrity as a topic. However, conventional KEPLMs simply neglect such linkages, and treat those sentences independently, which will lead to both *insufficient entity interaction* and *biased (relation) word semantics*. Take the sentence *"She released Crazy in Love and Baby Boy."* for example, if ignoring its topic entity Beyoncé, KEPLMs can no longer rely on this sentence to capture the interaction between Crazy in Love and Beyoncé. Moreover, the semantics of word "released" will be biased since it is between the above pair of entities, not between Crazy in Love and a common word "she".

Based on the above discussions, topic entities are indeed important to sentence semantics in Wikipedia-like textual corpus. However, they are non-trivial to model with simple revisions to existing KEPLMs. Readers may wonder whether they can be treated similarly as mentioned entities, *e.g.,* assign them certain position embeddings. However, we note that it is impractical since most topic entities do not explicitly appear in the sentences like mentioned entities[2]. Readers may also think of using a co-reference resolution model to replace words like "she" with the topic model's mention before feeding them to KEPLMs. However, embedding such a model to KEPLMs will not only introduce resolution noise, but also be insufficient to cover cases where topic entity information can clarify the local semantics of non-pronoun positions, *e.g.,* entity mentions with ambiguous names[3].

In this paper, we seek to fix the systematic neglect of topic entities in existing KEPLM efforts. We develop KÉPLET, a general **K**nowledge-**É**nhanced **P**re-trained **L**anguag**E** model with **T**opic entity awareness, which applies to most KEPLMs.

---

[2]In our initial analysis, for top-500K popular Wikipedia pages, only 6% of these pages mention the topic entity.

[3]"Crazy in Love" can refer to not only Beyoncé's song, but also an album of Itzy, a Korean girl group.

To exploit mature KEPLMs as its base while filling in the gap that topic entities do not have explicit positions, KÉPLET features a topic entity fusion module. To integrate topic entities into KEPLMs, KÉPLET first identifies potential *fusing positions* in sentences where topic entities can clarify local ambiguity including but not limited to co-references, through a gated neural network function (Hochreiter and Schmidhuber, 1996). It then *fuse* the topic entity features into the hidden representations of those positions in a soft manner. Finally, KÉPLET trains the fusion module with the base KEPLM in an end-to-end manner, through a specially designed topic-entity-aware *contrastive loss*. To validate the generality and effectiveness of KÉPLET, we conducted comprehensive experiments based on two representative KEPLMs, *i.e.,* LUKE (Yamada et al., 2020) and ERNIE (Zhang et al., 2019), on entity-centric benchmarks. The results demonstrate that KÉPLET consistently improves the performance of these two KEPLMs across the tasks.

We summarize our contributions as follows: **1)** We identify the systematic neglect of topic entities in existing KEPLM efforts. **2)** We propose KÉPLET with a novel topic entity fusion module and a topic-entity-aware loss, forcing existing KEPLMs to fully exploit entity knowledge in Wikipedia-like corpus. **3)** KÉPLET achieves the best performance among existing KEPLMs on several entity-centric benchmarks.

## 2 Preliminaries

In this section, we define two types of entities essential in Wikipedia-based KEPLM pre-training, *i.e., mentioned entities* and *topic entities*. We also give an overview of how conventional KEPLMs enhance PLMs by incorporating mentioned entities, but neglect topic entities.

### 2.1 Notations and Definitions

**Mentioned Entities.** On a Wikipedia page, each sentence $S$ consists of a sequence of *tokens* $W = \{w_i\}$ and hyperlinks under some tokens, linking to other Wikipedia pages and forming *mentioned entities*. For example, in Fig. 1, the last sentence with tokens $W = [She, released, Crazy, in, Love]$ mentions a song entity Crazy in Love with the last three words. In this paper, we denote mentioned entities in a sentence by $E = \{e_i\}$, where each $e_i$ has information about both the entity (*e.g.,* Wikipedia URL of Crazy in Love) and the position of the mention

span in the sentence.

**Topic Entities.** Besides mentioned entities, every Wikipedia sentence $S$ is also associated with another important entity, the topic entity (denoted by $e_t$), which is the entity of the page where the sentence is from. Topic entities are usually identified by the page URLs and indicated by the pages titles. For example, for the aforementioned sentence in Fig. 1, the topic entity $e_t$ is Beyoncé. Although not necessarily mentioning the topic entity $e_t$, all sentences on $e_t$'s page are usually around the topic of discussing all aspects of $e_t$.

## 2.2 PLMs, KEPLMs, and Neglect of Topic Entities

Based on the above notations, every Wikipedia sentence $S$ is essentially a tuple $S = \langle e_t, W, E \rangle$. On large-scale textual corpus, PLMs (Radford et al., 2018; Devlin et al., 2019; Liu et al., 2019) work on token sequences $W$ by training *transformer*-style encoders via specially designed losses $\mathcal{L}_{PLM}$ (*e.g., masked language modeling*).

**KEPLMs.** On top of PLMs, KEPLMs like LUKE (Yamada et al., 2020) and ERNIE (Zhang et al., 2019) make a further step by utilizing the mentioned entity information in $E$ (Safavi and Koutra, 2021). Specifically, they extend the encoders to also generate contextualized vectors for the entity mentions, and design entity disambiguation losses $\mathcal{L}_{ED}$ to make those vectors capable of predicating the topic entities. To avoid undermining the lexical and synthetic information of the PLMs, KEPLMs pre-train by jointly optimizing its loss and the conventional PLM losses, *i.e.,*

$$\mathcal{L}_{KEPLM} = \mathcal{L}_{PLM} + \mathcal{L}_{Aux.} \tag{1}$$

From a holistic point of view, KEPLMs aim to use $\mathcal{L}_{KEPLM}$ to pre-train a language model $\mathcal{LM}$ that can infer contextualized hidden representations for both tokens $W$ and mentioned entities $E$ in a Wikipedia-like sentence, *i.e.,*

$$\mathcal{LM} : \langle e_t, W, E \rangle \rightarrow \langle \mathbf{H}_w, \mathbf{H}_e \rangle, \tag{2}$$
$$\mathcal{LM} : \langle W, E \rangle \rightarrow \langle \mathbf{H}_w, \mathbf{H}_e \rangle. \tag{3}$$

and fine-tune $\mathcal{LM}$ in entity-centric downstream tasks. Note that there is a special hidden vector $\mathbf{h}_{[CLS]}$ in $\mathbf{H}_w$, which is the conventional sentence representation and will be useful in various downstream tasks as well as this work.

**Neglect of Topic Entities.** Although more effective than PLMs in entity-centric tasks, conventional KEPLMs did not fully exploit the rich structure inside a Wikipedia corpora. By working on only $W$ and $E$ of a Wikipedia sentence $S = \langle e_t, W, E \rangle$, those KEPLMs neglect the linkage between $S$ and its topic entity $e_t$, leading to two *weaknesses* as follows: **1)** Besides transformer parameters and initial word vectors, KEPLMs also have a third type of parameters, *i.e.,* initial entity vectors, to keep information of entities seen on the pre-training corpora for both topic and mentioned entities. Neglecting $e_t$ of every sentence will deteriorate the learning of vectors for entities occurring more as topic entities but mentioned less. **2)** In a Wikipedia corpus, it is crucial for words representing relations, *e.g.,* "released" in *"She released 'Crazy in Love' "*, to learn a good initial representation for downstream tasks like relation classification. Neglecting $e_t$ of every sentence will cause relation words in $W$ to interact only with $E$ rather than both, losing their semantics that they characterize certain relations between $e_t$ and $E$.

In this paper, we improve KEPLMs by bringing such linkages back into the architecture and training of $\mathcal{LM}$ in Eq. 3, through which we make up for the above two weaknesses.

## 3 KÉPLET: Integrating Topic Entities in KEPLMs

So far, we have motivated the fusion of topic entities into KEPLMs. The task then boils down to finding proper places in KEPLM training to fuse *topic entities* information. For mentioned entities, existing KEPLMs (Yamada et al., 2020; Zhang et al., 2019; Liu et al., 2020; Lu et al., 2021; Xiong et al., 2020; Févry et al., 2020) fuse their information by considering them as special tokens spanning multiple positions in a sentence. However, topic entities are non-trivial to incorporate in a similar manner, as they do not directly appear in the sentence. Naively assigning them artificial positions and doing insertions/replacements will potentially break the semantics of the sentences, thus deteriorating the LM's training.

In this section, we detail how KÉPLET integrates topic entities. KÉPLET features two modules, *fusing position identification* and *entity feature fusion*, and we customize the transformer encoders to accommodate the two modules. Besides, it employs a topic-entity-aware *contrastive learning* loss. Fig. 2

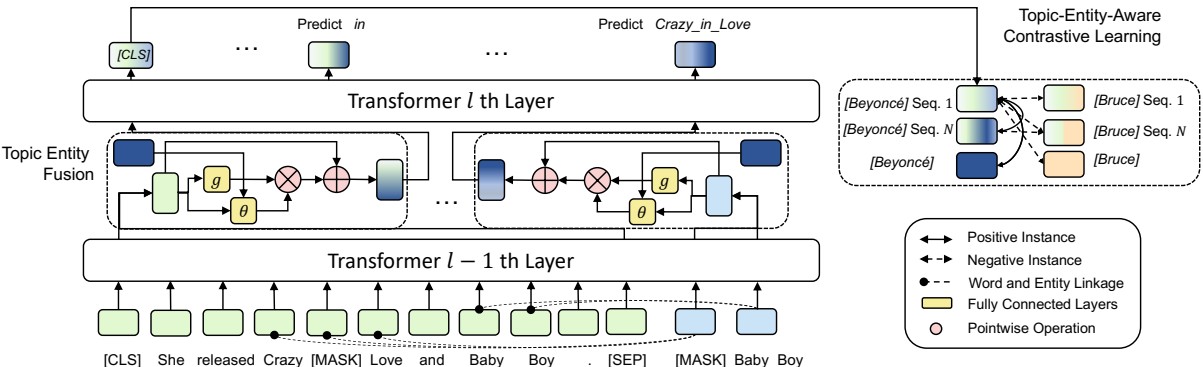

Figure 2: Illustration of KÉPLET with input *"She released Crazy in Love and Baby Boy"*. The mentioned entities Crazy in Love and Baby Boy are linked with the mentioning words. The middle left part is topic entity fusion module interleaved between transformer layers. The model is trained to predict the masked words and entities as in the top left. Topic-entity-aware contrastive learning is used to enforce the topic entity information by minimizing the distance among sentences' representations under Beyoncé and hidden representations of Beyoncé while enlarging the distance from sentences of other topic entities, *e.g.,* Bruce.

illustrates the architecture of KÉPLET.

### 3.1 Entity Fusing Position Identification

In conventional KEPLMs, the encoders consist of multiple layers of transformers, each of which depends on the contextualized hidden representations of words and mentioned entities from the previous layer. Let $\mathbf{H}_S = \text{CONCAT}(\mathbf{H}_w; \mathbf{H}_e)$ be all hidden representations for a Wikipedia sentence output by layer $(l-1)$, and let $\mathbf{h}^{(i)}$ be the $i$-th column of $\mathbf{H}_S$. From the perspective of preserving the semantics of topic entities $e_t$, we want to find proper positions $i$ to alter the hidden vectors $\mathbf{h}^{(i)}$ with information of $e_t$ before they are sent to the next layer, so that $\mathbf{H}_S$ still carries $e_t$'s semantics even if it is used without knowing what $e_t$ is.

One ideal type of such positions would be pronoun co-references referring to the topic entity, *e.g.,* word *"She"* in *"She released 'Crazy in Love' "*. They are reasonable places to fuse $e_t$'s information, since the ambiguous semantics of those positions will be clarified after the fusion. However, we note that co-references are not the only instances of such positions. For example, as word mentions, "Crazy in Love" can refer to not only Beyoncé's song, but also an album of Itzy, a Korean girl group. Fusing Beyoncé's information to the three words will thus benefit the representation of this ambiguous span. To this end, we resort to an end to end approach of identifying fusing positions instead to embed a co-reference resolution model (Ye et al., 2020) in KÉPLET, for the latter cannot cover all cases and is also error-prone due

to potentially imperfect model. For each position $i$, we compute $g_p^{(i)}$ indicating the necessity of fusing $e_t$ to position $i$, *i.e.,*

$$g_p^{(i)} = \sigma(F_p(\mathbf{h}^{(i)})). \qquad (4)$$

Here $F_p$ is a *fully connected layer* and $\sigma$ is the sigmoid activation function. To qualitatively evaluate the effectiveness of this module, we did a case study in § 5.5.

### 3.2 Entity Feature Fusion

With fusing positions softly identified by $g_p^{(i)}$, the entity feature fusion module aims to inject $\mathbf{e}_t$, *i.e.,* information of the topic entity $e_t$ to the corresponding $\mathbf{h}^{(i)}$. Inspired by the idea of *Adapters* (Houlsby et al., 2019) interleaved between the transformer layers, we apply an adapter function on $\mathbf{h}^{(i)}$ and $\mathbf{e}_t$ to create a fused representation $\hat{\mathbf{h}}^{(i)}$. We then follow the gating mechanism to softly combine $\hat{\mathbf{h}}^{(i)}$ and the original $\mathbf{h}^{(i)}$ for an updated input $\tilde{\mathbf{h}}^{(i)}$ to the next layer $l$, *i.e.,*

$$\hat{\mathbf{h}}^{(i)} = \text{Adapter}(\mathbf{h}^{(i)}, \mathbf{e}_t), \qquad (5)$$

$$\tilde{\mathbf{h}}^{(i)} = \text{LN}((1 - g_p^{(i)}) * \mathbf{h}^{(i)} + g_p^{(i)} * \hat{\mathbf{h}}^{(i)}). \qquad (6)$$

Here $*$ is multiplication and LN refers to *layer normalization*.

As for Adapter$(\cdot, \cdot)$, we had two implementations: *concatenation fusion* and *attention fusion*. We will compare them in the experiment section. **Concatenation Fusion.** This fusion approach uses a fully connected layer $F_t$ to transform the topic entity vector $\mathbf{e}_t$, then concatenates it with the hidden

representations $\mathbf{h}^{(i)}$ and feeds the result through another fully connected layer $F_c$ as:

$$\text{Adapter}(\mathbf{h}, \mathbf{e}) = F_c(\text{CONCAT}(\mathbf{h}; F_t(\mathbf{e}_t))) \quad (7)$$

Although the concatenation fusion is quite parameter efficient, it assumes that the words' and mentioned entities' hidden representation $\mathbf{h}$ as well as topic entity vectors $\mathbf{e}_t$ are from an unified feature space. This may be incorrect since the topic entity did not have the position information, while the words and mentioned entities had them.

**Attention Fusion.** To discern among topic entity, words, and mentioned entities, we propose the attention fusion. For each hidden vector $\mathbf{h}$, there is an attention fusion around it and the topic entity vector $\mathbf{e}_t$ as follows:

$$\text{Adapter}(\mathbf{h}, \mathbf{e}) = \text{softmax}\left(\frac{F_q(\mathbf{h})F_k(\mathbf{H})^\top}{\sqrt{d}}\right)F_v(\mathbf{H}) \quad (8)$$

$$\mathbf{H} = \text{CONCAT}(\mathbf{h}; \mathbf{e}_t). \quad (9)$$

Here $F_q$, $F_k$, $F_v$, are three fully-connected layers to get the query, key and value in the self-attention, and $d$ is the dimension size of hidden vectors $\mathbf{h}_i$.

Finally, we note that all fully connected layers in KÉPLET are layer-specific, *i.e.,* they do not share parameters across different layers in the transformer.

### 3.3 Topic-Entity-Aware Contrastive Loss

To enforce KÉPLET to really fuse the topic entity information into its output sentence and entity representations, we leverage the co-occurrences of sentences around the same $e_t$ as supervision signals. We design a novel contrastive learning loss $\mathcal{L}_t$, as shown in the right part of Fig. 2. The positive pairs are sentences' and topic entity's representation from the same topic entity, while the negative pairs are the sentences' and topic entities' representation from other topic entities in the minibatch.

Concretely, given output sentence representation $\mathbf{h}_{[CLS]}$ from token representations $\mathbf{H}_w$ of a Wikipedia sentence $S$ and its topic entity's representation $\mathbf{e}_t$, we compute the per-sentence loss $\mathcal{L}_S$ as follows. We denote $\Delta_S = \{\mathbf{h}_{[CLS]}, \mathbf{e}_t\}$, and all such vectors of Wikipedia sentences on $t$ by

$$\Delta_t = \bigcup_{S \text{ is on } t} \Delta_S. \quad (10)$$

For each $\mathbf{h} \in \Delta_S$, we draw a positive sample $\mathbf{h}^+$ from $\Delta_t$. We also draw negative samples $\Delta'$ from

vectors $\Delta_{t'}$ for a different topic entity $t'$. We then compute the contrastive learning loss as follow,

$$\mathcal{L}_S = -\sum_{\mathbf{h} \in \Delta_S} \sum_{\mathbf{h}^+ \in \Delta_t} \log \frac{e^{\text{sim}(\mathbf{h},\mathbf{h}^+)/\tau}}{\sum_{\mathbf{h}' \in \{\mathbf{h}^+\} \cup \Delta'} e^{\text{sim}(\mathbf{h},\mathbf{h}')/\tau}}. \quad (11)$$

Here *sim* is the cosine similarity between vectors and $\tau$ is a temperature hyperparameter. Finally, the overall objective function of KÉPLET is the sum of $\mathcal{L}_S$ across the corpora and $\mathcal{L}_{KEPLM}$ in Eq. 1.

## 4 Experiment Settings

We evaluate the effectiveness of KÉPLET on extensive entity-centric tasks: entity typing, relationship classification, named entity recognition and extractive QA. To ensure a fair comparison, we follow the experiment settings of previous work (Yamada et al., 2020; Zhang et al., 2019; Févry et al., 2020).

### 4.1 Baseline Methods

We compare KÉPLET with the vanilla PLMs: 1) BERT (Devlin et al., 2019), 2) RoBERTa (Liu et al., 2019), and KEPLMs: 3) KEPLER (Wang et al., 2021b), which utilizes the additional knowledge embedding loss to enhance the factual triplets from the knowledge graph for the PLM; 4) K-Adapter (Wang et al., 2021a), which did not explicitly model the entities and adopted the factual knowledge into the external adapter. 5) ERNIE (Zhang et al., 2019), which injects the mentioned entities' static embedding from KB and distinct the mentioned entity from the negative sampled entities; 6) LUKE (Yamada et al., 2020), which contains a separated entity embedding and word embedding, and utilizes the auxiliary masked entity prediction besides the masked token prediction to optimize the model.

### 4.2 Implementation Details

The pretraining Wikipedia corpus is the same as the LUKE (Yamada et al., 2020)[4], and we follow the same data preprocessing steps (Yamada et al., 2020) to extract entities from hyperlinks. For each entity, we assign a unique entity ID. The pre-training of KÉPLET starts from LUKE (Yamada et al., 2020) and ERNIE (Zhang et al., 2019), and is optimized for 4.9K steps (1 epoch). The temperature $\tau$ for contrastive learning is set to 0.07. The masking entity rate and masked entity rate are set to 60%. The optimizer for the pre-training is AdamW

---

[4]https://archive.org/download/enwiki-20181220

| Models | Open Entity $F_1$ | TACRED $F_1$ | SQuAD 1.1 $F_1$ | CoNLL-2003 $F_1$ |
|---|---|---|---|---|
| -Con. | 76.00 | 70.53 | 91.44 | 93.09 |
| -Atten. | **76.38** | **70.90** | **92.19** | **93.64** |

Table 1: Results of different feature fusion modules on LUKE-base+KÉPLET.

and is warmed up for 2.5K steps with a learning rate as 1e-5. During the downstream tasks' training and evaluation, the topic entity fusion module will be discarded. This is because there is no topic entity for downstream inputs and the knowledge loss for topic entity, and mentioned entity has already been complemented in the KEPLMs during pre-training. For all these baseline methods, we fine-tune their checkpoints on the same hardware and package settings like KÉPLET. This may cause the performance gap between our reproduction and their previously reported results. The hyperparameter settings are described in Appendix B.

### 4.3 Entity-Centric Tasks

The fine-tuning of entity centric tasks cannot simply take the hidden representation of "[CLS]" to represent the whole sentence. These tasks require special procedures to better represent the entities inside the sentence. Readers can refer to (Yamada et al., 2020; Zhang et al., 2019) for detailed fine-tuning procedures. The summaries and evaluation metrics for these entity centric tasks are as follows:
**Entity Typing** is to predict the types of an entity given the entity mention and the contextual sentence around the entity mention. Following the previous experiment setting of (Zhang et al., 2019; Yamada et al., 2020), we use the Open Entity dataset (Choi et al., 2018) and only consider nine popular entity types. We report the precision, recall and micro-$F_1$ scores, and use the micro-$F_1$ score for comparison.
**Relation Classification** is to classify the correlation between two entities. The input data includes two entity mentions and contextual information around these two entities. We utilize TACRED dataset (Zhang et al., 2017) and also report the micro-$F_1$ score for the comparison.
**Named Entity Recognition** is to identify the entity from the given sentence. We utilize CoNLL-2003 (Sang and Meulder, 2003) dataset and report the span-level $F_1$ scores.
**Extractive Question Answering** is to answer a question by extracting text span from a given passage. We utilize SQuAD1.1 (Rajpurkar et al., 2016) and report the exact match (EM) and token-level $F_1$ on the development dataset.

## 5 Experiment Results

In this section, we aim to answer the following experimental questions: **EQ1**. Which feature fusion method can better integrate the topic entity with KEPLMs? **EQ2**. Can topic entity fusion improve the performance of KEPLMs on downstream tasks? and **EQ3**. How should the topic entity fusion module of KÉPLET be added to KEPLMs to achieve optimal performance?

### 5.1 Comparing Entity Feature Fusion Approaches

To answer **EQ1**, we conduct the experiment on entity typing and relation extraction tasks and make comparison between KÉPLET-Con and KÉPLET-Atten. As the results are shown in Tab. 1, we can observe that KÉPLET-Atten achieves better performance than KÉPLET-Con in all the tasks. This indicates there may exist the unnecessary information or noise from the topic entity towards the words and mentioned entities, so it is important to selectively to do the feature fusion. Based on this observation, in the following sections, we will only report better feature fusion model: KÉPLET-Atten.

### 5.2 Effectiveness of Topic Entity Fusion

To answer **EQ2**, we compare the performance of KÉPLET among the PLMs and KEPLMs. We report the experiment results in Tab. 2. Note that in the literature, different KEPLMs may be initialized with different base PLMs, *e.g.,* ERNIE with BERT-base but KEPLER with RoBERTa-base. Therefore, we conduct three groups of comparisons, each corresponding one of the three possible base PLMs (*i.e.,*, BERT-base, RoBERTa-base, and RoBERTa-large) as well as the baseline KEPLMs basing on it. KÉPLET is implemented on the best performing baseline KEPLMs, *i.e.,* ERNIE or LUKE, whose performance is reported in the same group. In general, we have the following observations:
**Effectiveness of KEPLMs in entity-centric tasks.** We observe that in most cases KEPLMs achieve better performance compared with their initialized language models. This is because KEPLMs can better capture the entity-level factual knowledge besides the syntax based word co-occurrence information.

| Models | Open Entity | | | TACRED | SQuAD 1.1 | | CoNLL-2003 | | |
|---|---|---|---|---|---|---|---|---|---|
| | Prec. | Rec. | $F_1$ | $F_1$ | EM | $F_1$ | Prec. | Rec. | $F_1$ |
| BERT-base | 76.37 | 70.96 | 73.56 | 66.00 | 80.90 | 88.20 | 91.28 | 87.45 | 89.32 |
| ERNIE | 78.42 | **72.90** | 75.56 | 67.97 | - | - | - | - | - |
| ERNIE+KÉPLET | **79.85** | 72.01 | **75.72** | **70.18** | - | - | - | - | - |
| RoBERTa-base | **80.62** | 71.43 | 75.74 | 67.95 | 85.18 | 91.49 | 93.24 | 92.49 | 92.86 |
| KEPLER | 76.78 | 72.43 | 74.54 | 70.70 | - | - | - | - | - |
| LUKE-base | 79.70 | 72.37 | 75.86 | 70.30 | 85.18 | 91.88 | 92.81 | 93.27 | 93.04 |
| LUKE-base+KÉPLET | 78.39 | **74.47** | **76.38** | **70.90** | **85.66** | **92.19** | **93.48** | **93.80** | **93.64** |
| RoBERTa-large | 79.36 | 73.90 | 76.53 | 71.73 | 87.92 | 94.16 | 91.23 | 92.89 | 92.37 |
| K-Adapter | 79.30 | **75.84** | 77.53 | 71.89 | - | - | - | - | - |
| LUKE-large | 79.53 | 75.11 | 77.26 | 71.81 | 88.17 | 94.29 | **94.27** | **94.39** | **94.33** |
| LUKE-large+KÉPLET | **79.98** | 75.32 | **77.58** | **72.33** | **88.41** | **94.47** | 94.03 | 94.10 | 94.07 |

Table 2: Results on entity-centric tasks, *i.e.,* entity typing (Open Entity), relationship extraction (TACRED), extractive question answering (SQuAD1.1), and named entity recognition (CoNLL-2003). Bold text indicates best performance and underlined text indicates the second-best performance. We mark "-" where the corresponding baseline was not previously applied on this task.

**Effectiveness of advanced KEPLMs.** We observe that LUKE achieves better performance than ERNIE on all the entity-centric tasks. This is because the LUKE can learn the better contextual embedding of mentioned entities, while the ERNIE utilizes the static entity embedding.

**Effectiveness of KÉPLET over KEPLMs.** By considering the topic entity knowledge, KÉPLET gets considerable improvement on most tasks, achieving the best or second-best performance within the same comparison group. Specifically, on TA-CRED, KÉPLET gets 0.6% performance improvement compared with LUKE-base. This not only indicates the importance of integrating the topic entity into KEPLMs pretraining, but also represent the generalization of KÉPLET. Overall, KÉPLET achieves the best performance on all the datasets and metrics. KÉPLET brings the consistent performance improvement over the backbone KEPLMs.

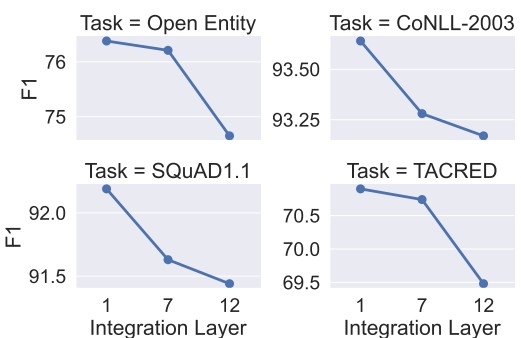

Figure 3: Parameter analysis of topic entity fusion layer $l$ on LUKE-base+KÉPLET.

## 5.3 Parameter Analysis

To address **EQ3**, we vary where and how many layers of topic entity fusion modules to be added

| # Layers | Open Entity $F_1$ | TACRED $F_1$ | SQuAD 1.1 $F_1$ | CoNLL-2003 $F_1$ |
|---|---|---|---|---|
| 1 | **76.38** | **70.90** | **92.19** | **93.64** |
| 2 | 76.21 | 70.80 | 91.60 | 93.12 |
| 3 | 75.66 | 70.01 | 91.58 | 93.31 |

Table 3: Results of different number of topic entity fusion module layers.

to the LUKE-base model, which has 12 transformer layers. We first conduct a parameter analysis of adding the fusion module between layers $l$ and $l-1$" layer $l$. As shown in Fig. 3, we can observe that fusion at lower layer achieves better performance than higher layer. Specifically, $l = 1$ achieves the best performance compared with $l = 7 and 12$. Since the lower layers contain more general information and the higher layers contain more task-specific information (Rogers et al., 2021), the topic entity can bring the general and comprehensive knowledge for the mentioned entities and word semantics.

In addition, we also evaluate the effectiveness of inserting different numbers of topic entity fusion modules in lower layers. We make the comparison among the single, double and triple topic entity fusion modules on LUKE-base backbone. As the results shown in Tab. 3, single position fusion layer achieves the best performance. This observation represents the efficiency of feature fusion module that can bring enough topic entity knowledge by only doing one time feature fusion.

## 5.4 Ablation Study on Topic Entity Fusion

To delve deeper into the impact of our entity fusing position identification (FPI) and entity feature fusion (FF) components, we performed an abla-

tion study. This allowed us to empirically measure the contribution of each module. Given computational constraints, we limited our evaluation to the `LUKE-base` backbone model. As illustrated in Tab. 4, omitting any particular component adversely affected the model's performance. These findings bolster our claim that arbitrary position assignments for topic entity fusion can harm overall effectiveness.

| # Layers | Open Entity F$_1$ | TACRED F$_1$ | SQuAD 1.1 F$_1$ | CoNLL-2003 F$_1$ |
|---|---|---|---|---|
| KÉPLET | **76.38** | **72.33** | **94.47** | **94.07** |
| *w/o* FPI | 74.96 | 70.97 | 91.71 | 93.40 |
| *w/o* FF | 75.93 | 70.22 | 91.77 | 93.41 |

Table 4: Ablation study of KÉPLET on `LUKE-base`. *w/o* stands for without, `FPI` is fusion position identification and `FF` is feature fusion.

## 5.5 Case Study of Fusion Position Identification

We aim to highlight the sentence segments that KÉPLET perceives as probable integration points. As delineated in Tab. 5, we visualize the rankings of integrated positions derived from the $g_p^{(i)}$ rank of the LUKE-base backbone. Notably, the model frequently targets pronouns such as "he" and named entities like `Farrukhsiyar` and `Western text-type`. This pattern aligns with the theoretical expectation: introducing a topic entity can refine sentence syntax and augment the representation of the cited entities. Such findings corroborate that `Model` primarily integrates broad topic entity knowledge rather than task-centric details.

## 6 Computational Overhead Discussion

Our model, trained on Knowledge-Enhanced `PLMs` (`KEPLMs`) for a single epoch, requires 128 hours on an A10 GPU—far less than the 11,520 hours needed for a KEPLM. Assuming each minibatch holds $M$ documents of $C$ tokens each, the added computational burden over KEPLMs is computed as $M \times C + M \times C \times C + M \times (M-1)$. Here, $M \times C$ represents the topic entity fusion position identification, $M \times C \times C$ corresponds to attention-based entity fusion, and $M \times (M-1)$ pertains to the Topic-Entity-Aware Contrastive Loss. This overhead is minimal as both $M$ and $C$ are small constants. Furthermore, during inference, KÉPLET bypasses topic entity position and fusion identification, keeping its computational demands in line with standard knowledge-enhanced language models.

## 7 Related Work

We group the related KEPLMs works based on what context information is used when encoding sentences with entity knowledge (*i.e.,* mentions), which are depending on *sentential context only* or additional *non-sentential context*.

**Sentential Context Only KEPLMs.** Most KEPLMs pre-trains by encoding sentences with mentioned entities and aligning the encoded spans with their ground truth entities. `ERNIE` (Zhang et al., 2019) integrates static entity embeddings into their PLM with a novel fusion layer to fuse spans' representations with entities'. `LUKE` (Yamada et al., 2020) treats both words and entities as tokens but uses different embeddings, self-attention layers, and masked token/entity prediction heads. It averages position embeddings of words in a span as the position embedding of the entity. Different from `ERNIE` and `LUKE`, which require mention spans to be available, `KnowBERT` (Liu et al., 2020) and `EAE` (Févry et al., 2020) incorporate entity linking modules to identify the spans, which are trained in an end-to-end manner. To deal with entity co-references in sentences, `CorefBERT` (Ye et al., 2020) and `TOME` (de Jong et al., 2021) propose extending the alignments to not only explicit entity mentions but also implicit co-reference spans. `WKLM` (Xiong et al., 2020) pre-trains by distinguishing the ground truth entity's embeddings from those of randomly corrupted ones. `K-Adaptor` (Wang et al., 2021a) utilizes plug-in adapters (Houlsby et al., 2019) to inject factual and linguistic knowledge without updating the LMs.

**Non-Sentential Context KEPLMs.** While KEPLMs depending only on sentential context have shown efficacy, other works argue that the alignment between mention spans and entities can be more effectively done if the mention spans are enriched with additional context outside the current sentence. `GRAPHCACHE`(Wang et al., 2022) constructs non-sentential context by building a heterogeneous graph with sentence and property nodes, and their interaction edges. Besides textual context, there are also works (Wang et al., 2021b; Liu et al., 2020; He et al., 2020; Su et al., 2021; Sun et al., 2020; Zhang et al., 2021) utilizing external knowledge graphs (KGs) as more informative non-sentential context. For example, `KEPLER` (Wang et al., 2021b) injects mentioned entities' descriptions and KG facts into its PLM. `K-BERT` (Liu et al., 2020) fuses KG facts into sentences by con-

| Topic Entity | Words and Mentioned Entities |
|---|---|
| Hugh Bonneville | Hugh Williams Jared November ACP ), professionally as Bon ne ville is English crew is best 408 Robert Craw ley in the series CRA ( 2010 2015 surging he a Globe Award two Emmy rockets SHARE . was born in Tues puck folder Oscar , to ight who SAY father who a . **He** was educated ributes clothing College at Sher borne School , independent in D orset . Following secondary Bon ne ville at Corpus Christ i , , PASS predictions Web ber Douglas of . Dire Cambridge a exorc beer in theology that **he** tended to do bath than academic work Bon ne ville is of the Youth Theatre . Bon ne ville 's professional IRC appearance was the Open Air avalanche . In 1987 Theatre foreground he the Royal 1991 , where 1966 's Ham let ( 1992 1993 ). **He** in railroad of ' T is 's a Wh ore , K ast ril and later in The Alchemist . , Bon ne ville made television debut , as Bon ne ville . **His film** was 1994 's Mary Shelley opot and Kenneth roles - nat ured b umbling characters like ( 1999 Mr Park Gap BBC series , Armored ( 2000 and played more villain ous characters , leading the dom ine ering Hen leigh Der onda ( 2002 ) and resilience seized risky ) In Again he played hours poet L arkin . Iris ), young Bay ley Orth his by and B AFTA In Mountain Luther in The Man ridicule . Bon ne ville works in radio MIA #Joachim Murat# **#Primetime Emmy Award for Outstanding Lead Actor in a Drama Series#** #Paddington# #National Youth Theatre# #Stafford# #2014 FIFA World Cup# #Single-elimination tournament# **#Gia Long# #Farrukhsiyar#** #Bridget Jones's Diary# #Cousin# #Juicy Couture# #Nationalism# #British Rail Class 08# |
| Lucius Valerius Messalla Thrasea Priscus | Val er ius Thr ase a ( d ied c . was a Roman active the Grind televised of Sept imus He was cons ul 196 inate colleague G ai us . at Dating tally was a of g ens . It is V ip stan us , who may have Statements a design atus but died **he** to the consulate so , Thr ase a gent il icum descent from the down from the ERA , held the office Carr dime nefarious ( or ) in Rome around AD 198 . listed been a trivial upper Pub l ius Sept im ius Get a , the brother and rival of emperor . He became one of of earliest Car ac alla 's autical Get a initions Christian has speculated that slight unarmed brightly Pr isc us married possibly sought Caribbean , a speech close Brom the future . is Pr isc us save a son , Lucius Mess alla Ap oll in aris , in ronic . In ge , and in Roman Empire , AD 193 - 284 ( 2011 Mess alla Pa etus , Lucius #Łobez# #Ukrainians# #Bernie Sanders# #Roman aqueduct# #Budapest# #Andrés de Santa Cruz# **#Women's Tennis Association# #Western text-type# #Ukrainians#** |

Table 5: Examples of fusing positions. Top 50 important words are highlighted. The color saturation indicates the importance. Bold and colored words are either pronouns relevant to the topic entity and the phrases wrapped by # are mentioned entities.

structing a sentence tree. Compared with textual context, KG-based context often requires specially processed and linked knowledge from Wikipedia or WikiData, where noises could be introduced in those processes. In our work, we explore topic entities as a unique non-sentential context that is directly available in the Wikipedia structure without additional processing.

**Contrastive Learning in KEPLMs.** Orthogonal to various types of context, multiple objectives or tasks have been introduced by previous studies to pre-train those KEPLMs. Most of those attempts fall in the contrastive learning setting to minimize distances between positive pairs while enlarging those between negative ones (Hadsell et al., 2006; Gao et al., 2021; Chen et al., 2020; Meng et al., 2021). Therefore, the construction of positive pairs and negative pairs is crucial in this setting. In UCTopic (Li et al., 2022), positive pairs are sentences that share the same entity mentions, while negative ones are from different mentions. EASE (Nishikawa et al., 2022) extends UCTopic by incorporating entity representations into positive pairs. LinkBERT (Yasunaga et al., 2022) introduces a sentence relation classification objective

to make sentence representations capable of predicting sentences' relations. Unlike those works, we construct positive and negative pairs with the unique topic entity information to guide the semantic learning of entities and relation words.

## 8 Conclusion

We propose KÉPLET to address the oversight of topic entities in KEPLM learning. KÉPLET performs topic entity fusion by identifying the fusing positions and fusing topic entity features through concatenation- or attention-based fusion. Furthermore, we design a new pre-training task, *i.e.,* topic-entity-aware contrastive learning, for better topic entity fusion. Experiment results on several entity-centric tasks prove the effectiveness of KÉPLET. Potential directions of the future work include: **1)** Adopting KÉPLET to multi-language setting. Since topic entities are language invariant (Nishikawa et al., 2022), we can expand the topic-entity-aware contrastive learning for the same topic entity under different languages; **2)** Applying KÉPLET to domain-specific knowledge-centric tasks like fact checking and fake news detection.

## 9 Limitations

Since our method has incorporated the topic entity into language model pretraining, it requires an entity-rich pretraining dataset. The dataset's layout should have topic entity and mentioned entities. The demand for special pretraining dataset is a clear limitation of our method. In addition, our pretraining stage require many computational resources (eight A10 cards for up to 16 hours, so totally 128 GPU hours). This is because we want to inject the previously neglected topic entity knowledge from Wikipedia into KEPLMs. This will require us to do the further pretraining on the whole Wikipedia dataset. However, given the performance improvements brought by our method on many entity-centric tasks, the additional computational cost is totally worthy.

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

## A   Details of Pretraining

Following the work of LUKE(Yamada et al., 2020), we utilize the same pretraining dataset: December 2018 version of Wikipedia[5] and hyperparameters, except learning rate (1e-6). We separate different input sequences from same Wikipedia page by 512 words. To provide enough positive pairs for Topic-Entity-Aware Contrastive Loss, we modify the code of data minibatch generation for pretraining. Specifically, we firstly combine consecutive elements of the pretraining dataset into batches, then do the shuffling and lastly split the batch into multiple elements. The TensorFlow-style code block is like follows:

```
# LUKE's implementation
data = data.shuffle()
# Our implementation
data = data.batch(2).shuffle().unbatch()
```

## B   Details for Fine-Tuning

Following the hyper-parameter setting from LUKE (Yamada et al., 2020), we conduct the hyper-parameter search on all the datasets except SQuAD1.1. We use grid search to find the best model based on the validation performance. The metrics for model selection are reported in § 4.3. All in all, we use the following search space:

- learning rate 1e-5, 2e-5, 5e-5, 5e-6
- batch size: 4, 8, 16, 32, 64
- number of training epochs: 2, 3, 5

## C   Computational Overhead

Our model is trained on existing Knowledge-Enhanced `PLMs` (`KEPLMs`) for just one epoch, which typically takes 128 hours on A10 GPU. This is significantly less than the 11,520 hours required to train a KEPLM itself. To provide a more detailed breakdown, let's assume each minibatch contains $M$ documents, with each document having $C$ tokens. The additional computational cost of ours over KEPLMs is $M \times C + M \times C \times C + M \times (M-1)$, where $M \times C$ is the cost for topic entity fusion position identification and $M \times C \times C$ is the cost for attention-based entity fusion and $M \times (M-1)$ is the Topic-Entity-Aware Contrastive Loss. It is relatively modest compared to the previous method because $M$ and $C$ are small constant values. In addition, at inference time, KÉPLET avoids the need for topic entity position identification and fusion during inference. This ensures that the computational expense remains comparable to the standard knowledge-enhanced language model.

---

[5]https://github.com/studio-ousia/luke/issues/112