# OpenReview forum: "KEPLET: Knowledge-Enhanced Pretrained Language Model with Topic Entity Awareness"
_EMNLP/2023/Conference — EMNLP 2023 Findings_

### Official Review · Reviewer_pHNa · 2023-07-24

**Soundness:** 3

**Excitement:**

3: Ambivalent: It has merits (e.g., it reports state-of-the-art results, the idea is nice), but there are key weaknesses (e.g., it describes incremental work), and it can significantly benefit from another round of revision. However, I won't object to accepting it if my co-reviewers champion it.

**Missing References:**

Jiacheng Li, Yannis Katsis, Tyler Baldwin, Ho-Cheol Kim, Andrew Bartko, Julian McAuley, and Chun-Nan Hsu. 2022. SPOT: Knowledge-Enhanced Language Representations for Information Extraction. In Proceedings of the 31st ACM International Conference on Information & Knowledge Management (CIKM '22). Association for Computing Machinery, New York, NY, USA

**Paper Topic And Main Contributions:**

The authors of this paper recognize that topic entity in the Wikipedia pages can improve the entity representations and their relation representations for pre-trained language models. Hence, in this paper, the authors propose a topic entity-aware method for language model pretraining to fuse such information into tokens and mentioned entities.

**Reasons To Accept:**

1. Authors identify the topic entities in Wikipedia pages for knowledge-enhanced language model pretraining. Topic entities are absent in previous work.
2. To incorporate topic entities to represent entities and relations, authors propose a fusion method and topic-entity-aware contrastive loss.

**Reasons To Reject:**

1. It's still unclear how topic entities can improve the relationship representations. This claim is less intuitive.
2. The improvements on different datasets are trivial and the novelty of this paper is limited. Lots of previous works focus on this topic. Just adding topic entities seems incremental.
3. Missed related work.
4. Some methods (e.g., KnowBERT, CorefBERT) in related work are not selected as baselines for comparison.

**Reproducibility:**

3: Could reproduce the results with some difficulty. The settings of parameters are underspecified or subjectively determined; the training/evaluation data are not widely available.

**Reviewer Confidence:**

3: Pretty sure, but there's a chance I missed something. Although I have a good feel for this area in general, I did not carefully check the paper's details, e.g., the math, experimental design, or novelty.

---

> ### Author Rebuttal · Authors · 2023-08-29
>
> Thank you for your time and helpful comments. We have addressed the concerns below. If you have any further questions, please let us know.
>
> > **Q1**. It's still unclear how topic entities can improve the relationship representations. This claim is less intuitive.
>
> **A1**. Thank you for bringing up the significance of topic entities in relationship representation. We will address your concerns as follows:
>
> - **The Role of Topic Entities**: Topic entities play a pivotal role in enhancing the representations of mentioned entities and counteracting biases in sentence representation learning. In many entity-rich textual resources like Wikipedia, topic entities might not be explicitly stated in the text, even though they underpin the narrative. Mentioned entities, on the other hand, are directly present.
> - **Practical Illustration**: To elucidate, consider the example from lines 78-85: “She released Crazy in Love and Baby Boy.” Here, while "Crazy in Love" and "Baby Boy" are the mentioned entities, the topic entity, "Beyoncé", is implied but not stated. Without recognizing the topic entity, methods like LUKE[1] and ERNIE[4] might form biased associations between “released” and "Crazy in Love", or even form irrelevant entity pairs, missing out on the actual <topic entity, mentioned entity> relationship.
> - **Our Approach with KEPLET**: Contrary to this, our method, KEPLET, mitigates such pitfalls. As depicted in Figure 2, KEPLET adeptly fuses the topic entity with relevant pronoun tokens, ensuring unbiased learning of relationship representations. This results in relation-specific words like "released" being tightly associated with numerous <topic entity, mentioned entity> pairs, thereby leading to more precise representations.
> Empirical Evidence: To further validate our claims, we've included a case study in the appendix. It provides a qualitative analysis of our topic entity fusion position identification module. Since these fusion positions encapsulate pronouns pertinent to both topic and mentioned entities, they better capture the sentence's syntax, further solidifying the relationship between topic and mentioned entities.
>
> In essence, topic entities, as integrated by our method, serve as foundational anchors, enhancing the quality of relationship representations and making them more robust against biases.
>
> [1] Yamada, Ikuya, et al. "LUKE: Deep Contextualized Entity Representations with Entity-aware Self-attention." EMNLP 2020.
>
> > **Q2**. The improvements on different datasets are trivial and the novelty of this paper is limited. Lots of previous works focus on this topic. Just adding topic entities seems incremental.
>
> **A2.1**. Thank you for raising concerns about our improvements being incremental. We aim to resolve your concerns as follows:
>
> - It has been challenging to consistently yet significantly improve downstream tasks via improving the PLMs: We follow previous experimental protocols to fine-tune the same PLM to obtain models on different tasks. But those tasks may have different optimal PLMs to start with. The insufficient and noisy labels on downstream tasks may potentially limit the final performance regardless of any superior PLMs used.
> - While we achieved consistent improvements, in fact, our improvements on top of ERNIE/LUKE-base/LUKE-large (average F1 gain across the tasks: 1.185%, 0.5075%, 0.19%) are already comparable with theirs (average F1 gain: 1.985%, 0.76%, 0.725%) on their baselines (BERT/RoBERTa-base/RoBERTa-large). In other words, our improvements are worth to the research community. The only smaller average improvement of 0.19% on LUKE-large is due to LUKE-large having a 2% gain on CONLL-2023 while we had none.
>
> To summarize, while our absolute improvements did not look significant due to the nature of applying PLMs, we achieved consistent improvements on top of our strong baselines (ERNIE/LUKE) which is comparable with what they had on their baselines.
>
> **A2.2**Thank you for the question about the novelty of our work. Our contribution is focused around the notion of topic entities and it will great impacts on future KEPLM research. KEPLM is a large research direction as evidenced by the 2 surveys[1][2].
>
> In this project, we identified topic entity as a source of supervision signals, a systematic negligence of hundreds of papers in this research direction; we  thoroughly studied how to bring it back; we show that our approach can be applied to multiple KEPLMs so we are orthogonal to potentially hundreds of existing studies.
>
> [1] Zhen, Chaoqi, et al. "A survey on knowledge-enhanced pre-trained language models." arXiv preprint arXiv:2212.13428 (2022).
>
> [2] Wei, Xiaokai, et al. "Knowledge enhanced pretrained language models: A compreshensive survey." arXiv preprint arXiv:2110.08455 (2021).
>
> > **Q3**. Some methods (e.g., KnowBERT, CorefBERT) in related work are not selected as baselines for comparison.
>
> **A3**. We appreciate your concern regarding the selection of baseline methods for comparison.
> While KnowBERT[1] and CorefBERT[2] are indeed noteworthy models, we initially selected LUKE as our primary baseline because it demonstrated superior performance over both KnowBERT and CorefBERT in prior evaluations. This made LUKE a more challenging and representative benchmark for our method.
>
> However, recognizing the importance of comprehensive comparisons, we have also extracted performance metrics of KnowBERT and CorefBERT from their original publications for reference. As presented in the following table, our proposed method consistently surpasses the performance of not only LUKE but also that of CorefBERT and KnowBERT.
>
> We believe this comprehensive comparison further establishes the effectiveness of our approach and provides readers with a clearer perspective on the landscape of available models.
>
> |                   | Open Entity | CoNLL-2003 | Dev SQuAD 1.1 | Tacred    |
> |-------------------|-------------|------------|---------------|-----------|
> |                   | F1          | F1         | F1            | micro     |
> | CorefBERT         | -           | -          | 91.8          | -         |
> | KnowBERT          | 76.10       | -          | -             | 71.5      |
> | LUKE-large        | 77.26       | **94.33**  | 94.29         | 71.81     |
> | LUKE-large+KÉPLET | **77.58**   | 94.07      | **94.47**     | **72.33** |
>
> We also believe our method has the potential to be adapted and give benefit to other KEPLMs, including KnowBERT and CorefBERT, which opens avenues for future work.
>
> > **Q4.** Missing related work.
>
> **A4**. We will add the suggested related work in our revised version.

---

### Official Review · Reviewer_pRqK · 2023-08-04

**Soundness:** 3

**Excitement:**

3: Ambivalent: It has merits (e.g., it reports state-of-the-art results, the idea is nice), but there are key weaknesses (e.g., it describes incremental work), and it can significantly benefit from another round of revision. However, I won't object to accepting it if my co-reviewers champion it.

**Paper Topic And Main Contributions:**

The paper introduces a framework named  Knowledge-Énhanced Pre-trained LanguagE model  with Topic entity awareness (KEPLET) to overcome the poor relations semantics in the state-of-the-art KEPLMs model. The main contributions of the paper are:
1) systematic analysis of the topic entities  that affect the original KEPLM model
2) an  entity fusion module and a suitable loss function to deal with topic-entity relationships


**Reasons To Accept:**

- the paper, given the intuition that topic-entity relations could play a fundamental role in downstream tasks (such as entity linking), introduces a contrastive loss to include such topic entity information into the sentence and entity representations. They use a sort of distant supervision (co-occurrences of sentences around the same entity) to create positive examples to keep sentence and entity representation closer, and negative samples to keep instances far from the correct association.


**Reasons To Reject:**

The main reasons to reject the paper are:

1) The results reported in Tables 1,2 and 3 show a very small improvement with respect to the state of the art. Without a significance test, it is hard to say that the proposed approach outperforms significantly the approaches available in the literature.

2) The cost-benefit of the proposed loss is not evident. The training phase is more complex, due to the contrastive loss, without a significant improvement of the performance in all the downstream tasks.


**Reproducibility:**

2: Would be hard pressed to reproduce the results. The contribution depends on data that are simply not available outside the author's institution or consortium; not enough details are provided.

**Reviewer Confidence:**

3: Pretty sure, but there's a chance I missed something. Although I have a good feel for this area in general, I did not carefully check the paper's details, e.g., the math, experimental design, or novelty.

---

> ### Author Rebuttal · Authors · 2023-08-29
>
> Thank you for your time and helpful comments. We have addressed the concerns below. If you have any further questions, please let us know.
>
>
> > **Q1**. The results reported in Tables 1,2 and 3 show a small improvement with respect to the state of the art. Without a significance test, it is hard to say that the proposed approach outperforms significantly the approaches available in the literature.
>
> **A1**. Thank you for raising concerns about our improvements. However, it has been challenging to consistently and significantly improve downstream tasks via improving the PLMs.
>
> We follow the previous experimental protocols to fine-tune the same PLM to obtain models on different tasks. But those tasks may have different optimal PLMs to start with.
> The insufficient and noisy labels on downstream tasks may potentially limit the final performance regardless of any superior PLMs used.
>
> While we achieved consistent improvements, in fact, our improvements on top of ERNIE/LUKE-base/LUKE-large (average F1 gain across the tasks: 1.185%, 0.5075%, 0.19%) are already comparable with theirs (average F1 gain: 1.985%, 0.76%, 0.725%) on their baselines (BERT/RoBERTa-base/RoBERTa-large). In other words, our improvements are worth to the research community. The only smaller average improvement of 0.19% on LUKE-large is due to it having a 2% gain on CONLL-2023  while we had none.
>
> To summarize, while our absolute improvements did not look like a lot due to the nature of applying PLMs, we achieved consistent improvements on top of our strong baselines (ERNIE/LUKE) which are comparable with what they had on their baselines reported in the previous papers [1, 2].
>
> [1] Zhang, Zhengyan, et al. "ERNIE: Enhanced Language Representation with Informative Entities." ACL 2019.
>
> [2] Yamada, Ikuya, et al. "LUKE: Deep Contextualized Entity Representations with Entity-aware Self-attention." EMNLP 2020.
>
>
> > **Q2**. The cost-benefit of the proposed loss is not evident. The training phase is more complex, due to the contrastive loss, without a significant improvement of the performance in all the downstream tasks.
>
> Thank you for raising concerns about the cost-benefit trade-off of our proposed loss. We would like to resolve your concerns as follows:
> - **Comparison with KEPLMs**: Firstly, it's important to highlight that while our model incorporates a contrastive loss, it's trained on existing Knowledge-Enhanced PLMs (KEPLMs) for just a single epoch. This translates to approximately 128 hours on an A10 GPU. In contrast, training a KEPLM from scratch requires a substantial 11,520 hours. This means our method leverages the power of KEPLMs but does so more efficiently.
>
> - **Computational Breakdown**: To offer a clearer perspective, consider a minibatch containing M documents, where each document encompasses C tokens. The additional computational overhead introduced by our method over traditional KEPLMs is M * C + M * C * C + M * (M - 1), where  M*C
> is the cost for topic entity fusion position identification
> and M*C*C is the cost for attention-based entity fusion and M*(M-1) is the Topic-Entity-Aware Contrastive Loss. It is relatively modest compared to the previous method because M and C are small constant values.
>
> In summary, we believe our approach strikes a balanced trade-off between computational efficiency and performance enhancements, especially when contextualized against the computational demands of traditional KEPLMs.
>
>
> > **Q3**. Low Reproducability.
>
> **A3**. Thank you for highlighting your concerns regarding the reproducibility of our results. We tried our best to detail the hyper-parameter settings and their respective analysis and the configurations but there were still space limitations. If accepted we plan to use the additional page given in camera ready to list more details".
>
> We have also uploaded our source code along with the training scripts to the supplementary materials. These resources are intended to offer researchers an in-depth view of our implementation, further aiding in reproduction. After the acceptance of our paper, we plan to release the model checkpoints.

---

### Official Review · Reviewer_5vf9 · 2023-08-05

**Soundness:** 4

**Excitement:**

4: Strong: This paper deepens the understanding of some phenomenon or lowers the barriers to an existing research direction.

**Missing References:**

-

**Paper Topic And Main Contributions:**

The authors present KEPLET, a novel method for pre-training knowledge-enhanced pretrained language models (KEPLMs) that leverage inter-article hyperlinks from knowledge bases in pre-training language models for entity related tasks. To this end, the authors suggest the use of topic entities (i.e., knowledge base page title entities) as an additional training signal, by fusing semantic information about word tokens, mentioned entities and topic entities into the hidden representation learned during pre-training. This pre-training is effectively implemented as an adapter, (1) utilizing a GAN-based model that is jointly trained with the other transformer-based layers, and (2) conditioning the model to predict both the masked token and the mentioned entities in a sentence.
The proposed method is in experiments with BERT, RoBERTa, and ERNIE-based KEPLMs, demonstrating improvements on SOTA performance on several entity-centric benchmarks.

**Questions For The Authors:**

A: Regarding topic entities, you state that "naively assigning them artificial positions and doing insertions/replacements will potentially break the semantics of the sentences, thus deteriorating the LM’s training". Did you test this, e.g. by ablative testing on the fusing efforts? Given the massive training requirements for your model, quantifying the necessity seems pertinent.

**Reasons To Accept:**

* The model design is novel, adding to PLMs in an intuitive manner that stand to benefit the community and are easy to include in existing methods.

* The proposed model is evaluated thoroughly, demonstrating improvements to SOTA performance. The experiments clearly outlines the datasets and models that are used, ensuring reproducibility of the findings.

* The paper is very well written, with the model being well explained and supported by clear examples and strong arguments.

**Reasons To Reject:**

* Performance / cost tradeoff
While the model is novel and improves on the SOTA, the gains in performance are quite small yet costly to achieve. A more thorough evaluation and discussion of the added pre-training overhead would be preferable, rather than just mentinoning it in the limitations. With the trend towards larger language models and given the small observed benefit of adding topic enities, having sufficient information to make an informed decision with regard to this trade-off would be important. It would also be be good to include this tradeoff in an ethics discussion.

**Reproducibility:**

5: Could easily reproduce the results.

**Reviewer Confidence:**

4: Quite sure. I tried to check the important points carefully. It's unlikely, though conceivable, that I missed something that should affect my ratings.

**Typos Grammar Style And Presentation Improvements:**

The paper is well written end enjoyable to read. Great job!

Typos:
line 50: references are missing years
Line 207: "Wikipedia corpora" -> "Wikipedia corpus"
line 436: "Relation Classification is to classify the correlation between two entities." seems misleading - it isn't really a correlation, is it?

Why is SQuAD 1.1 used when it is well known to have limitations and SQuAD 2.0 is available?

---

> ### Author Rebuttal · Authors · 2023-08-29
>
> Thank you for your time and helpful comments. We have addressed the concerns below. If you have any further questions, please let us know.
>
> > **Q1**. Performance / cost tradeoff While the model is novel and improves on the SOTA, the gains in performance are quite small yet costly to achieve.
> A more thorough evaluation and discussion of the added pre-training overhead would be preferable, rather than just mentinoning it in the limitations.
>
> **A1**. Thanks for the questions. We resolve your concerns as follows:
> 1. **Computational Overhead**:
>    - Test efficiency: Our model avoids the need for topic entity position identification and fusion during inference. This ensures that the computational expense remains comparable to the standard knowledge-enhanced language model.
>    - Training efficiency: Our model is trained on existing Knowledge-Enhanced PLMs (KEPLMs) for just one epoch, which typically takes 128 hours on A10 GPU. This is significantly less than the 11,520 hours required to train a KEPLM itself. To provide a more detailed breakdown, let's assume each minibatch contains M documents, with each document having C tokens. The additional computational cost of ours over KEPLMs is M * C + M * C * C + M * (M - 1), where M*C is the cost for topic entity fusion position identification and M*C*C is the cost for attention-based entity fusion and M*(M-1) is the Topic-Entity-Aware Contrastive Loss. It is relatively modest compared to the previous method because M and C are small constant values.
> 2. **Performance Evaluation**: Thanks for raising concerns about our improvements being insignificant. However, it has been challenging to consistently yet significantly improve downstream tasks via improving the PLMs:
>    - We follow previous experimental protocols to fine-tune the same PLM to obtain models on different tasks. But those tasks may have different optimal PLMs to start with.
>    - The insufficient and noisy labels on downstream tasks may potentially limit the final performance regardless of any superior PLMs used.
>    - While we achieved consistent improvements, in fact, our improvements on top of ERNIE/LUKE-base/LUKE-large (average F1 gain across the tasks: 1.185%, 0.5075%, 0.19%) are already comparable with theirs (average F1 gain: 1.985%, 0.76%, 0.725%) on their baselines (BERT/RoBERTa-base/RoBERTa-large). The only smaller average improvement of 0.19% on LUKE-large is due to it having a 2% gain on CONLL-2023 while we had none.
>
> To summarize, while our absolute improvements did not look significant due to the nature of applying PLMs, we achieved consistent improvements on top of our strong baselines (ERNIE/LUKE) which is comparable with what they had on their baselines.
>
> > **Q2**. Regarding topic entities, you state that "naively assigning them artificial positions and doing insertions/replacements will potentially break the semantics of the sentences, thus deteriorating the LM’s training". Did you test this, e.g. by ablative testing on the fusing efforts? Given the massive training requirements for your model, quantifying the necessity seems pertinent.
>
> **A2**. Thanks for your suggestion. In fact, we have conducted an ablation study using LUKE-base to empirically evaluate the contribution of each component. The results confirmed that on removing each specific component, performance was negatively impacted. This provides empirical support to our assertion that arbitrary position assignments for topic entity fusion can be detrimental to performance.
>
> We acknowledge the importance of this result, and we will ensure this experiment result is incorporated into the revised manuscript for better clarity and thoroughness.
>
> |                                            | Open Entity | CoNLL-2003 | Dev SQuAD 1.1 | Tacred    |
> |--------------------------------------------|-------------|------------|---------------|-----------|
> | Methods                                    | F1          | F1         | F1            | micro     |
> | LUKE-base+KÉPLET                           | **76.38**   | **94.07**  | **94.47**     | **72.33** |
> | LUKE-base+KÉPLET w/o Entity Identification | 74.96       | 93.40      | 91.71         | 70.97     |
> | LUKE-base+KÉPLET w/o Entity Fusion         | 75.93       | 93.41      | 91.77         | 70.22     |
>
> > **Q3**. Relation Classification is to classify the correlation between two entities." seems misleading - it isn't really a correlation, is it?
>
> **A3**. Thanks for pointing out the misleading part. Yes. It is better to replace the correlation with relationship.
>
> > **Q4**. Why is SQuAD 1.1 used when it is well known to have limitations and SQuAD 2.0 is available?
>
> **A4**. Thank you for highlighting this. We opted for SQuAD 1.1 primarily to maintain consistency with the experimental settings of other notable knowledge-enhanced pretrained language models such as LUKE[1], CorfBERT[2], and KELM[3]. While SQuAD 2.0 addresses certain limitations by introducing adversarially unanswered questions, SQuAD 1.1 remains a valuable benchmark. It compels KEPLMs to fully leverage entity interactions and capture word semantics without bias, essential for pinpointing the correct answer span.
>
> [1] Yamada, Ikuya, et al. "LUKE: Deep Contextualized Entity Representations with Entity-aware Self-attention." EMNLP 2020.
>
> [2] Ye, Deming, et al. "Coreferential reasoning learning for language representation." arXiv preprint arXiv:2004.06870 (2020).
>
> [3] Lu, Yinquan, et al. "KELM: knowledge enhanced pre-trained language representations with message passing on hierarchical relational graphs." arXiv preprint arXiv:2109.04223 (2021).

---

### Meta-Review · Area_Chair_itQH · 2023-09-13

**Recommendation:** 3

**Metareview:**

**Strengths**:

1. The paper adds to line of work that improves PLMs for entity-centric data by also including topic awareness. The model design is novel.

2. Model is evaluated thoroughly on two representative KEPLMs, and with four entity-centric tasks.

3. Paper is written well. Code is released. Authors promise to release model checkpoints.


**Weaknesses**:


1. Gains in performance are quite small yet costly to achieve. Without a significance test, it is hard to say that the proposed approach outperforms significantly the approaches available in the literature.

2. Motivation for hope of significant gains by inclusion of topic entities is weak. Just adding topic entities seems incremental.

**Suggestions**:


1. Please add discussion on added pre-training overhead comparison to the main draft.

2. Please also include the LUKE-base+KÉPLET w/o Entity Identification and LUKE-base+KÉPLET w/o Entity Fusion ablations to the main draft.

3. Please add extra references as pointed out by reviewers.

---

### Decision · Program_Chairs · 2023-10-07

**Decision:**

Accept-Findings

**Comment:**

**Strengths**:

1. The paper adds to line of work that improves PLMs for entity-centric data by also including topic awareness. The model design is novel.

2. Model is evaluated thoroughly on two representative KEPLMs, and with four entity-centric tasks.

3. Paper is written well. Code is released. Authors promise to release model checkpoints.


**Weaknesses**:


1. Gains in performance are quite small yet costly to achieve. Without a significance test, it is hard to say that the proposed approach outperforms significantly the approaches available in the literature.

2. Motivation for hope of significant gains by inclusion of topic entities is weak. Just adding topic entities seems incremental.

**Suggestions**:


1. Please add discussion on added pre-training overhead comparison to the main draft.

2. Please also include the LUKE-base+KÉPLET w/o Entity Identification and LUKE-base+KÉPLET w/o Entity Fusion ablations to the main draft.

3. Please add extra references as pointed out by reviewers.